# Lineage-specific changes in mitochondrial properties during neural stem cell differentiation

Rita Soares[1,2,3] , Diogo M Lourenço[1,2] , Isa F Mota[1,2,3], Ana M Sebastião[1,2], Sara Xapelli[1,2,*] , Vanessa A Morais[1,3,*]

**Neural stem cells (NSCs) reside in discrete regions of the adult mammalian brain where they can differentiate into neurons, astrocytes, and oligodendrocytes. Several studies suggest that mitochondria have a major role in regulating NSC fate. Here, we evaluated mitochondrial properties throughout NSC differentiation and in lineage-specific cells. For this, we used the neurosphere assay model to isolate, expand, and differentiate mouse subventricular zone postnatal NSCs. We found that the levels of proteins involved in mitochondrial fusion (Mitofusin [Mfn] 1 and Mfn 2) increased, whereas proteins involved in fission (dynamin-related protein 1 [DRP1]) decreased along differentiation. Importantly, changes in mitochondrial dynamics correlated with distinct patterns of mitochondrial morphology in each lineage. Particularly, we found that the number of branched and un-branched mitochondria increased during astroglial and neuronal differentiation, whereas the area occupied by mitochondrial structures significantly reduced with oligodendrocyte maturation. In addition, comparing the three lineages, neurons revealed to be the most energetically flexible, whereas astrocytes presented the highest ATP content. Our work identified putative mitochondrial targets to enhance lineage-directed differentiation of mouse subventricular zone–derived NSCs.**

## Introduction

Despite having been demonstrated in several studies since the 1960s that new neurons are continuously being generated in the adult brain contributing to neural plasticity (Altman, 1962; Nottebohm, 1985; Eriksson et al, 1998; Ernst & Frisén, 2015), the existence of this phenomenon in humans has been debated in other studies (Sorrells et al, 2018; Franjic et al, 2022). Importantly, it has been suggested that the differences found in these studies are because of several factors including sample collection and preparation. Despite the controversy, the vast majority of the studies support the existence of human hippocampal neurogenesis. To further understand this dynamic event, neural stem cell (NSC) biology is currently the subject of intense study. NSCs are multipotent cells characterized by their proliferative and self-renewal capacity throughout the life time of the host, and their ability to exit the cell cycle to initiate differentiation (Hall & Watt, 1989; Gage, 2000; Temple, 2001). NSCs are able to differentiate into the three neural-ectoderm–derived populations of the nervous system: neurons, astrocytes, and oligodendrocytes in defined processes termed neurogenesis, astrogliogenesis, and oligodendrogenesis, respectively (Reynolds & Weiss, 1992; Gage et al, 1998; Obernier & Alvarez-Buylla, 2019). In the adult rodent brain, NSCs mainly reside in discrete regions of the brain, identified as neurogenic niches (Obernier & Alvarez-Buylla, 2019). One of these regions is the subventricular zone (SVZ) located along the wall of the lateral ventricle and that continuously generates olfactory bulb interneurons but also oligodendrocytes under demyelinating conditions (Butti et al, 2019). The other niche is the subgranular zone in the hippocampal dentate gyrus (DG) which generates granule cells (Wu et al, 2015). The self-renewal and differentiation capacities of the NSCs have been extensively studied by in vitro techniques, where a single NSC can respond to growth factors generating neurospheres or monolayer colonies that can both differentiate upon growth factor withdrawal (Reynolds & Weiss, 1992; Palmer et al, 1999). Particularly, the neurosphere assay has been widely used not only because it provides a consistent and unlimited source of NSC but also because the heterogeneous composition of the neurospheres mimics the in vivo niches (Soares et al, 2020, 2021).

NSC fate, involving the decision to self-renew or differentiate, has been largely explored because of its importance for both tissue development and regeneration. Recent evidence has emerged suggesting that mitochondria is involved in NSC differentiation and lineage determination both in mice and humans (Khacho et al, 2016; Ramosaj et al, 2021; Döhla et al, 2022; Petrelli et al, 2023). Mitochondria are complex organelles involved in bioenergetics, signaling pathways, and cell death (Mitchell, 1961; Green & Kroemer, 2004). In addition, mitochondria are dynamic organelles. Mitochondrial biogenesis is the formation of de novo mitochondria

[1]Instituto de Medicina Molecular | João Lobo Antunes (iMM|JLA), Faculdade de Medicina, Universidade de Lisboa, Lisbon, Portugal   [2]Instituto de Farmacologia e Neurociências, Faculdade de Medicina, Universidade de Lisboa, Lisbon, Portugal   [3]Instituto de Biologia Molecular, Faculdade de Medicina, Universidade de Lisboa, Lisbon, Portugal

Correspondence: sxapelli@medicina.ulisboa.pt; vmorais@medicina.ulisboa.pt
*Sara Xapelli and Vanessa A Morais contributed equally to this work

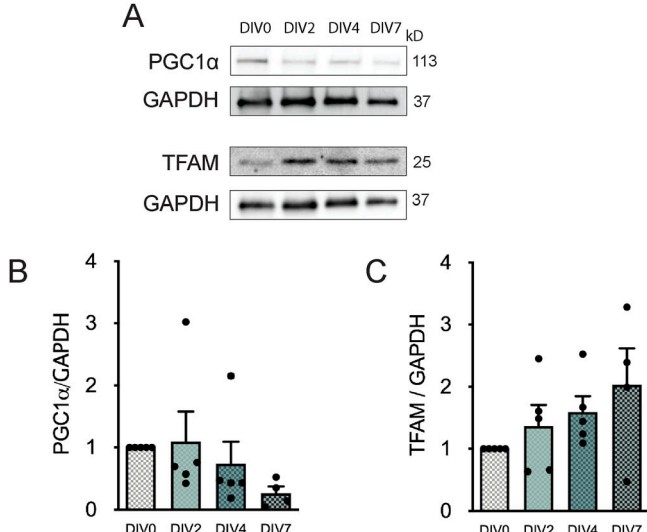

**Figure 1. The protein levels of PGC1α and mitochondrial transcription factor A (TFAM), master regulators of mitochondrial biogenesis, are unaltered with subventricular zone–derived neural stem cell differentiation.**
**(A)** Representative immunoblots depict the immunoreactive bands of PGC1α, TFAM, and GAPDH (loading control) proteins at DIV0, DIV2, DIV4, and DIV7 (left to right). **(B, C)** Quantitative analysis of PGC1α (B) and TFAM (C) protein levels normalized to GAPDH with differentiation of subventricular zone–derived neural stem cells. Data are presented as mean ± SEM (one-way ANOVA followed by Tukey's multiple comparison test).

from pre-existing ones, which requires mitochondrial DNA replication, transcription, and translation (Popov, 2020). Importantly, enhancement of mitochondrial biogenesis promotes the regenerative potential of the NSCs in aged mice (Stoll et al, 2015), whereas human NSC differentiation into motor neurons potentiates mitochondrial biogenesis (O'Brien et al, 2015). Although the role of mitochondrial biogenesis in glia cells is far from being understood, a recent report suggests that this process is required for the maturation of postnatal mouse astrocytes (Zehnder et al, 2021). Besides mitochondrial biogenesis, mitochondria continuously divide and fuse. Balanced mitochondrial dynamics and morphology are necessary to maintain a healthy pool of mitochondria within the cells, ensuring an appropriate mitochondrial function at the proper time and subcellular location to address the cellular requirements (Chen & Chan, 2004). Particularly, dysregulation of this process in embryonic human and mouse NSCs interferes with their self-renewal capacity and neurogenesis process (Steib et al, 2014; Khacho et al, 2016; Iwata et al, 2020). Similar findings were observed in the drosophila model in which depletion of mitochondrial fusion–related proteins cause a depletion in the type II neuroblasts (i.e., NSCs) pool, thereby leading to a reduction in the number of differentiated cells (Dubal et al, 2022). In addition, emerging evidence has shown that alterations of the mitochondrial morphology during hippocampal embryonic and adult neurogenesis are pivotal for decision-making regarding NSCs fate (Khacho et al, 2016; Beckervordersandforth et al, 2017). In addition, mitochondrial function is also fundamental for NSC fate determination as shown in mouse (Wani et al, 2022; Petrelli et al, 2023), human (Iwata et al, 2020), and drosophila models

(Homem et al, 2014; van den Ameele & Brand, 2019). Nonetheless, to date, the molecular mechanisms by which mitochondrial biogenesis, dynamics, and bioenergetics mediate postnatal NSC commitment are unknown. Therefore, the aim of this work was to assess how mitochondrial biogenesis and dynamics change along postnatal-SVZ–derived mouse NSC differentiation, exploring mitochondria morphology and bioenergetics in the distinct lineage-specific cells. Our results demonstrate that the levels of mitochondrial fusion– and fission–related proteins are significantly increased and decreased, respectively, along SVZ-derived NSC differentiation. Moreover, the mitochondrial number significantly increased during astroglial and neuronal differentiation, whereas the mitochondrial area significantly reduced along oligodendroglial maturation. Our data demonstrate that at later stages of NSC differentiation, cells are more reliant on oxidative phosphorylation (OXPHOS) and that neurons present higher energy flexibility.

# Results

## Mitochondrial biogenesis is not changed with NSC differentiation

The neurosphere assay, which was previously characterized by our group (Soares et al, 2020), is an appropriate model to evaluate the postnatal NSC fate as both stemness and multipotency properties of these cells can be determined in a heterogenic microenvironment that mimics the neurogenic regions. Notably, this model allows the simultaneous evaluation of postnatal differentiation into neurons, astrocytes, and oligodendrocytes (Weiss et al, 1996). To evaluate mitochondrial properties with NSC differentiation, NSCs were isolated from the SVZ. To guarantee a high yield of NSC population, we performed two passages, obtaining tertiary neurospheres. The cells within the neurospheres exhibit both stemness and proliferative capacity (Soares et al, 2020). SVZ neurospheres when plated under differentiative conditions (with the removal of growth factors from the medium) give rise to neuronal, oligodendroglial, and astroglial cells as shown in our previous work (Soares et al, 2020). In fact, throughout differentiation besides astrocytes, immature cells such as NSCs (SOX2+ cells), immature neurons (DCX+ cells), and oligodendrocyte precursor cells (OPCs) (PDGFRa+/NG2+ cells) are highly present at DIV2, whereas at DIV7, mature neurons (NeuN+ cells) and myelinating oligodendrocytes (MBP+ cells) are more expressed.

To investigate whether mitochondrial biogenesis is altered with postnatal NSC differentiation, we assessed the protein levels of the master regulators of this process: peroxisome proliferator–activated receptor-γ coactivator 1α (PGC1α) and the mitochondrial transcription factor A (TFAM). Once PGC1α is activated, this protein interacts with the nuclear respiration factors 1 and 2 (NRF-1 and NRF-2), leading to the expression of many mitochondrial genes and proteins that are needed for mitochondrial DNA replication and transcription, such as TFAM (Wu et al, 1999). Immunoblot analysis revealed no significant alteration in the protein levels of PGC1α (Figs 1A and B and S8A and B) and TFAM (Figs 1A and C and S8A and B) along NSC differentiation

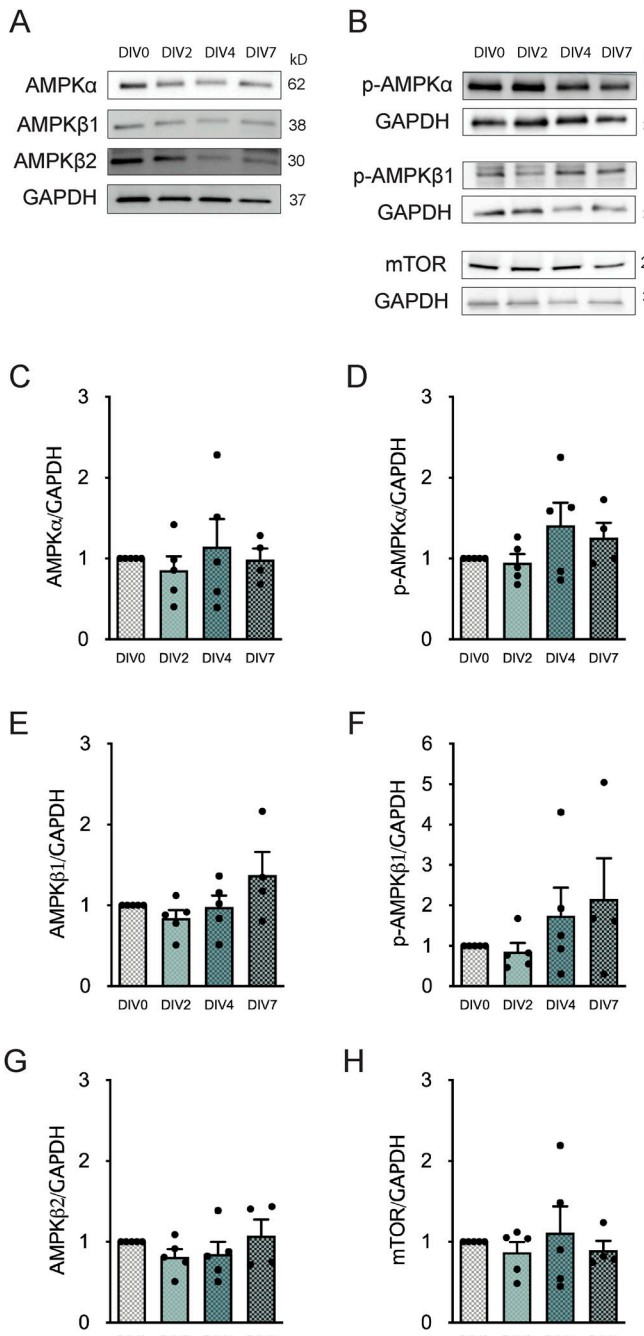

**Figure 2. Subventricular zone–derived neural stem cell differentiation does not impact on AMP-activated protein kinase (AMPK) and mechanistic target of rapamycin (mTOR) protein levels.**
**(A, B)** Representative immunoblots depict the immunoreactive bands of AMPKα, AMPKβ1, and AMPKβ2 (A) and p-AMPKα, p-AMPKβ1, and mTOR (B) at DIV0, DIV2, DIV4, and DIV7 (left to right). **(C, D, E, F, G, H)** Quantitative analysis of AMPKα (C), p-AMPKα (D), AMPKβ1 (E), p-AMPKβ1 (F), AMPKβ2 (G), and mTOR (H) protein levels normalized to GAPDH during subventricular zone–derived neural stem cell differentiation. Data are expressed as mean ± SEM (one-way ANOVA followed by Tukey's multiple comparison test).

(DIV0, DIV2, DIV4, and DIV7). Therefore, our data suggest that mitochondrial biogenesis might not have an impact in SVZ-derived NSC differentiation.

## NSC differentiation does not affect AMP-activated protein kinase (AMPK) protein levels

To further corroborate the previous findings indicating that mitochondrial biogenesis is not altered during SVZ-derived NSC differentiation, we explored the protein levels of indirect regulators of this process: AMPK and the mechanistic target of rapamycin (mTOR). AMPK is a heterotrimeric serine/threonine protein kinase with the capacity to phosphorylate PGC1α, through the reduction of ATP/AMP ratio in the cell (Bergeron et al, 2001; Reznick et al, 2007). Moreover, this kinase has a catalytic α-subunit and a scaffolding β subunit (β1 and β2). As for the master regulators, no changes were observed in the protein levels of AMPKα (Fig 2A and C), AMPKβ1 (Fig 2A and E), and AMPKβ2 (Fig 2A and G). In addition, no significant alterations were observed in phospho-AMPK (p-AMPKα) and p-AMPKβ1, the respective activated forms of AMPKα and AMPKβ1 (Fig 2B, D, and F). Also, no significant differences were observed in the p-AMPKα/ AMPKα (Fig S1A) and p-AMPKβ1/AMPKβ1 (Fig S1B) ratios. Hence, these results demonstrate that the overall protein levels of AMPK and its activated forms do not change throughout SVZ-derived NSC differentiation. Finally, mTOR another indirect regulator of mitochondrial biogenesis (Morita et al, 2013; Knobloch & Jessberger, 2017) also did not reveal significant changes in protein levels with NSC differentiation (Fig 2H). Overall, these findings further suggest that NSC differentiation does not affect mitochondrial biogenesis-related protein levels.

## Mitofusins 1 and 2 increase, whereas DRP1 decreases, during NSC differentiation

To disclose whether NSC differentiation and maturation affect mitochondrial dynamics, the protein levels of the outer membrane fusion proteins Mitofusin 1 (Mfn1) and Mfn2 (Rojo et al, 2002) and the inner membrane fusion protein OPA1 (Cipolat et al, 2004) were evaluated in SVZ cells (Figure S9A-D. Interestingly, Mfn2 protein levels significantly increased with NSC differentiation, reaching a maximum level of 3.457 ± 0.3969-fold (n = 4–5, P < 0.001) in cells at DIV7 (Fig S2B). In contrast, no significant alterations were observed in Mfn1 (Fig S2A) and in the OPA1 full-length form (upper band) and cleaved form (lower band) (Fig S2C and D). To further understand the role of NSC commitment in mitochondrial dynamics, we also explored mitochondrial fission by evaluating the protein levels of the dynamin-related protein (DRP1) (Smirnova et al, 2001). Noteworthily, SVZ cell differentiation induced a marked decrease of DRP1 to 20% ± 2.85% of its initial levels (DIV7: 0.2018 ± 0.02854, n = 4–5, P < 0.0001) (Fig S2E).

To determine whether these differences observed in the protein levels of mitochondrial fusion–related proteins were dependent on alterations in mitochondrial mass during NSC differentiation, the mitochondrial mass was assessed by quantifying HSP60 protein levels, a mitochondrial matrix protein (Green & Kroemer, 2004). As HSP60 levels have no major change along cell differentiation (Figs 3 and S9), Mfn1/2 and OPA1 protein levels were normalized to HSP60. Notably, a significant increase in Mfn1 protein levels was observed in SVZ cells only at DIV7 (2.287 ± 0.3801, n = 4–5, P < 0.05) (Fig 3C). In respect to Mfn2 levels, the significant alterations were maintained

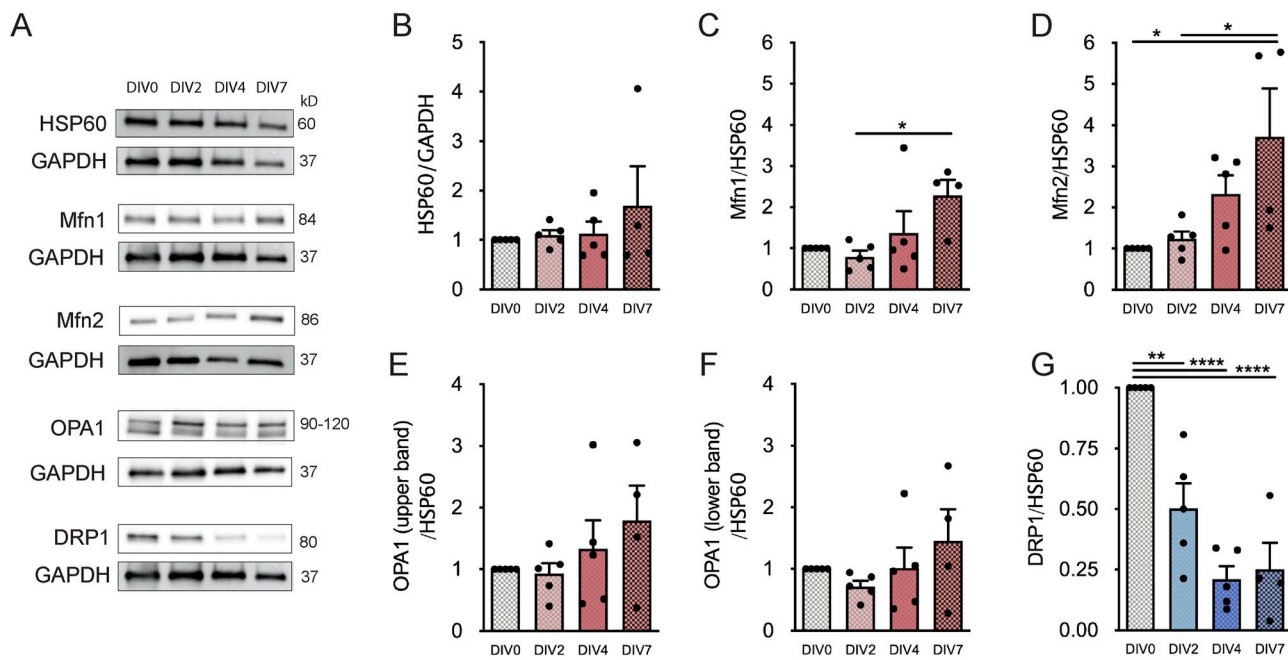

**Figure 3. Mitochondrial fusion– and fission–related protein levels significantly change along neural stem cell differentiation.**
**(A)** Representative immunoblots depict the immunoreactive bands of all the proteins at DIV0, DIV2, DIV4, and DIV7 (left to right). **(B)** Analysis of HSP60 protein levels with neural stem cell differentiation at P0-2. (n = 4–5). **(C, D, E, F, G)** Quantitative analysis of Mfn1 (C), Mfn2 (D), OPA1 full-length form (E), OPA1 cleaved form (F), and DRP1 (G) normalized by HSP60 protein levels. Data are expressed as mean ± SEM (*P < 0.05, **P < 0.01, ****P < 0.0001, one-way ANOVA followed by Tukey's multiple comparison test).

(DIV2: 1.230 ± 0.1821, DIV7: 3.724 ± 1.161, n = 4–5, P < 0.05) (Fig 3D). No significant changes were observed in the OPA1 full-length and cleaved form (Fig 3E and F). Regarding DRP1, the fission-related protein continued to significantly decrease with cell differentiation reaching 25.10% ± 10.91% of the initial levels (DIV7: 0.2510 ± 0.1091, n = 4–5, P < 0.0001) (Fig 3G). Overall, our results suggest a role for mitochondrial fusion/fission machinery in SVZ-derived NSC differentiation.

### Each neural cell type displays a unique mitochondrial morphology profile

Because the mitochondrial network is remodeled by fusion and fission events, and based on the results obtained in Fig 3, we postulated that both undifferentiated and differentiated cells could present distinct mitochondrial morphologies that can be lineage-dependent and maturation degree–related, in SVZ cells. To further explore this hypothesis, mitochondrial morphology was studied within undifferentiated (Fig S3) and differentiated cells (Figs 4, S4, and S5) including astrocytes, immature and mature neurons, OPCs, and different stages of oligodendrocytes. By using the MiNA macro (Valente et al, 2017), we evaluated several morphometric parameters including the number of individuals (rods and unbranched puncta), the number of networks (branched structures), the average number of branches per network, the average length of the rods and branches, and the percentage of mitochondrial area (Fig 4A). Importantly, these parameters, when considered together, allow to draw conclusions about mitochondrial fragmentation and fusion/fission events. Interestingly, the number of mitochondria, individuals, and networks significantly

increased in NSCs (SOX2+ cells) from DIV0 to DIV2 (Fig S3B and C), during astroglial differentiation (GFAP+ cells) and in mature neurons (αtau+ cells) (Figs 4B and C and S4A and C). No significant changes were observed in immature neurons (DCX+ cells) from DIV2 to DIV4 regarding mitochondrial individuals (Figs 4B and S4B) and mitochondrial networks (Figs 4C and S4B), whereas in mature neurons, an increase is observed with differentiation. By contrast, no significant changes in the mitochondrial number were observed in OPCs (PDGFRα+/NG2+ cells) with differentiation (Figs 4B and C and S5A). Curiously, the number of mitochondria is lower in the more mature oligodendrocytes (MBP+ cells with complex branched structures) when compared with less mature (MBP+ cells with poorly branched structures) at both DIV4 and DIV7. However, the number of mitochondria was already low in mature oligodendrocytes at DIV4 and had similar values when comparing with mature oligodendrocytes at DIV7, suggesting that the number of mitochondria decreases to lower levels with oligodendrocyte maturation (Figs 4B and C and S5B). Interestingly, the more mature oligodendrocytes present a lower number of mitochondria when compared with astrocytes and neurons (Figs 4B and C, S4A–C, and S5B).

A significant reduction in the number of branches per network was observed in astrocytes from DIV4 to DIV7 and in both immature neurons and mature neurons (Figs 4D and S4A–C), whereas no alterations in NSCs (Fig S3D) and oligodendroglial cells (Figs 4D and S5A and B) were observed. Interestingly, in immature neurons, the mitochondria became more elongated as cells mature returning then to a less elongated phenotype in mature neurons (Figs 4E and S4B and C). In astrocytes, mitochondria become less elongated with differentiation but then

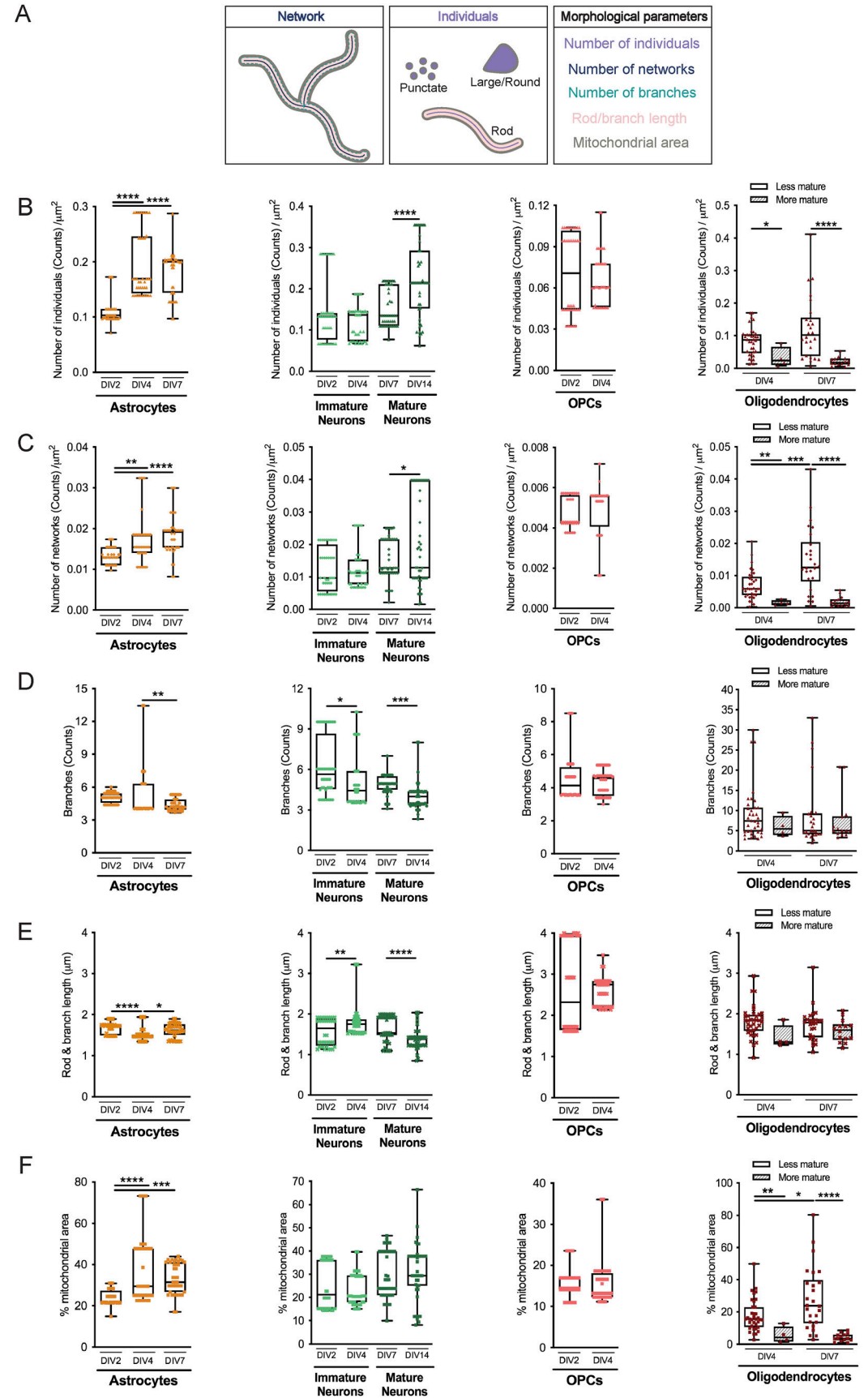

return to a more elongated morphology (Figs 4E and S4A). NSCs revealed a significant decrease in the mitochondrial length from DIV0 to DIV2 (Fig S3E). Moreover, this parameter remained unchanged in oligodendroglial cells (Figs 4E and S5A and B). Lastly, when we analyze the mitochondria area in the cells, we found a significant increase in astrocytes (Figs 4F and S4A) along differentiation. In addition, as oligodendrocytes undergo maturation, the mitochondrial area is reduced to approximately half (Fig 4F) and a perinuclear distribution of the mitochondria is observed (Fig S5B). Therefore, these robust alterations of mitochondrial morphology could suggest that mitochondrial dynamics could be important modulators of SVZ-derived NSC fate; however, further validation is required.

### NSC differentiation at the later stages requires higher levels of ATP production

Because mitochondrial dynamics and morphology are to a certain extent linked to mitochondrial bioenergetics, the dissimilarities observed in the different fates derived from NSCs might be because of differences in the bioenergetic profile. Therefore, respiratory assays were performed in control cells at different differentiation time-points (24, 96, and 96 h + 4 d in low EGF/FGF-2 medium). The extracellular acidification rate, a proxy for glycolysis, and the oxygen consumption rate (OCR) mediated by mitochondrial respiration were assessed by using the Mito Stress setup (Faria-Pereira et al, 2022). The energy map (Fig 5A) revealed that under stressed conditions (upon FCCP injection), these three controls do not present major changes. Nevertheless, we then evaluated the mitochondrial respiration rates (Fig S6A and B). No changes were observed in all the parameters (Fig 5B, D–G), with the exception of the ATP-linked respiration where a significant increase with differentiation and cell maturation from the 96 h to the 96 h + 4 d time-points (96 h: 66.30% ± 5.239% and 96 h + 4 d: 77.19% ± 2.925%, n = 6–7, P < 0.05) was observed (Fig 5C). Overall, these data suggest that throughout NSC differentiation, particularly at the later time-points, there was a more pronounced reduction in the % of OCR when mitochondrial ATP production was inhibited.

### Neurons are the most energetically flexible cells differentiated from the SVZ-derived NSCs, whereas oligodendrocytes are the least flexible

We then assessed the energy map of the sorted astrocytes, oligodendrocytes, and neurons (Fig 6A), showing that the neurons are more aerobic when exposed to stress condition in comparison with glial cells that present a more glycolytic profile.

In addition, neurons present significantly lower levels of basal respiration (astrocytes: 74.65% ± 1.616%, oligodendrocytes: 75.03% ± 2.347%, and neurons: 61.41% ± 4.538%, n = 4–9, P < 0.01) (Figs 6B and S7A) and proton leak (astrocytes: 25.35% ± 1.541%, oligodendrocytes: 24.49% ± 2.295%, and neurons: 15.05% ± 1.474%, n = 4–9, P < 0.01) (Fig 6D) in comparison with the glial cells. In addition to this, ATP-linked respiration did not change among the different cell types (Fig 6C). Neurons present a significantly higher maximal respiration in comparison with glial cells (astrocytes: 143.50% ± 6.526%, oligodendrocytes: 117.60% ± 9.755%, neurons: 184.80% ± 24.83%, n = 4–9; P < 0.05 and P < 0.01) (Figs 6E and S7B) and a significantly higher spare respiration in comparison with both astrocytes and oligodendrocytes (astrocytes: 68.82% ± 6.359%, oligodendrocytes: 42.61% ± 10.30%, neurons: 123.8% ± 26.31%, n = 4–9, P < 0.01 and P < 0.001) (Figs 6F and S7B), which suggests that cells that commit to the neuronal lineage present higher energy flexibility. Finally, significant differences were observed in non-mitochondrial respiration between neurons and glial cells (Fig 6G). Concluding, neurons are the most energetically flexible cells despite having the lowest basal respiration among the differentiated cells from the SVZ-derived NSCs.

### Astrocytes have a significantly higher ATP content when compared with oligodendrocytes and neurons

To assess if the changes observed upon ATP-linked production (Fig 5D) in 96 h + 4 d control comparing with the 96 h were reflected in total ATP content, the levels of ATP were assessed in the 24, 96, and 96 h + 3 d controls and in the sorted cells (astrocytes, oligodendrocytes, and neurons).

No significant differences were observed in the ATP content levels when comparing all control conditions (Fig 7A). Importantly, the fact that the ATP content is similar between the 96 h and the 96 h + 3 d (Fig 7A) and because the 96 h + 4 d presents a significantly higher ATP-linked to respiration (Fig 5D) suggests that at 96 h + 3 d, there could be a higher rate of ATP consumption or that at 96 h, there could be a higher rate of ATP production by other pathways. Interestingly, although no differences were observed in the ATP-linked respiration in the sorted cells (Fig 6D), we observe that astrocytes present a significantly higher ATP content (astrocytes: 3.488 ± 1.042 μM/μg protein, oligodendrocytes: 0.9669 ± 0.3219 μM/μg protein, neurons: 0.4758 ± 0.2264 μM/μg protein, n = 4, P < 0.01 and P < 0.001) (Fig 7B). Overall, the data of the ATP content together with the ATP-linked respiration give new insights about the ATP consumption and the ATP production by other pathway in some conditions.

**Figure 4. Mitochondrial morphology varies among neural lineages and with differentiation.**
**(A)** Illustrative image showing the color-coded morphological parameters evaluated. **(B, C, D, E, F)** Quantitative analysis of mitochondrial individuals (B), mitochondrial networks (C), number of branches per network (D), mitochondrial length (rods and branches) (E), and mitochondrial area (F), in the different neural lineages. Data are normalized by the cytoplasmic area. Data are represented as box plots, showing the median with interquartile range, maximum and minimum values. *P < 0.05, **P < 0.01, ***P < 0.001, ****P < 0.0001 by Student's test (oligodendrocyte precursor cells, immature and mature neurons), one-way ANOVA followed by Tukey's multiple comparison test (astrocytes), and Mann–Whitney test (oligodendrocytes). N = 40/40/40 and n = 6/7/13 (left to right on astrocytes' plot); N = 40/40/43/45 and n = 6/9/12/22 (left to right on neurons' plot); N = 40/40 and n = 5/7 (left to right on oligodendrocyte precursor cells' plot); N = 36/4/26/14 and n = 33/4/26/13 (left to right on oligodendrocytes' plot). N represents the number of biological replicates, and n corresponds to the number of ROIs analyzed.

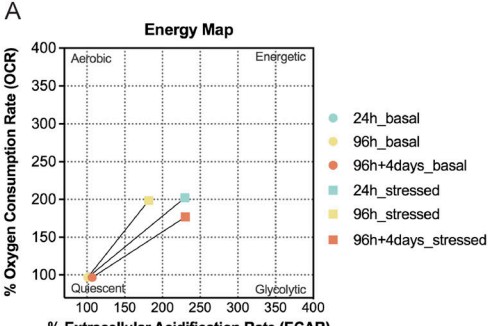

**Figure 5. Neural stem cell differentiation at the later stages requires higher levels of ATP production.**
**(A, B, C, D, E, F, G)** Energy map, (B) quantification of basal respiration, (C) ATP-linked respiration, (D) proton leak upon response to oligomycin, (E) maximal and (F) spare respiration upon response to FCCP, and (G) non-mitochondrial respiration upon response to rotenone and antimycin A. Data are presented as mean ± SEM (*$P < 0.05$, one-way ANOVA followed by Tukey's multiple comparison test).

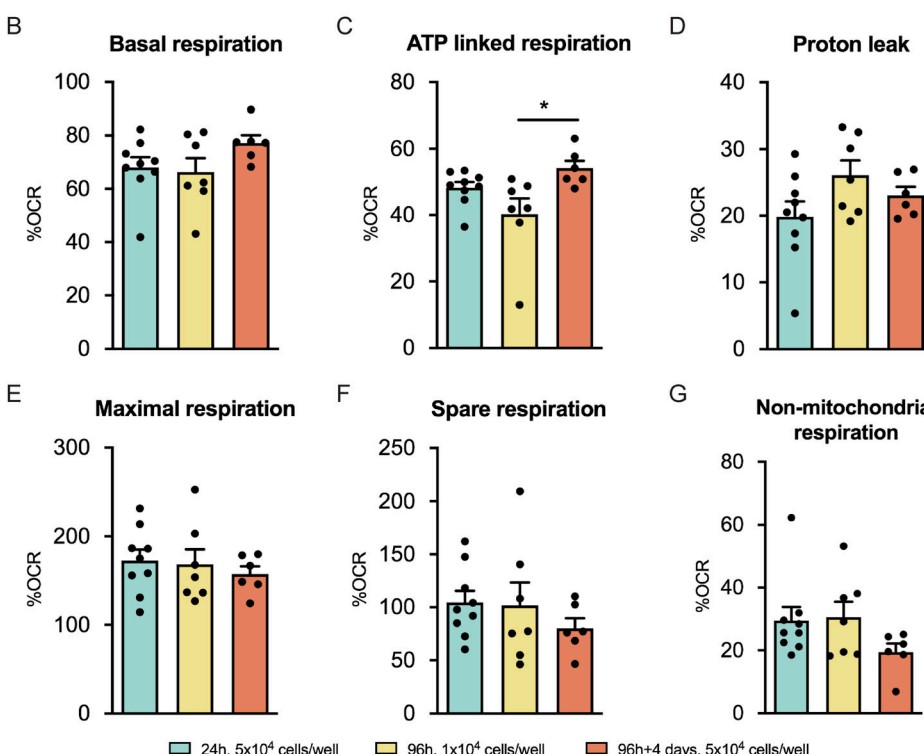

## Discussion

NSCs are valuable therapeutic targets, holding the capability to rebuild and restore tissue in the adult brain through the formation of new neural cells (Reynolds & Weiss, 1992; Gage et al, 1998; Obernier & Alvarez-Buylla, 2019). Here, we provide important evidence about the role of mitochondrial dynamics and metabolism toward postnatal mouse NSC commitment into a specific lineage, opening the avenue to potential approaches that impose NSC fate by modulating mitochondrial properties.

In our study, protein profiles of key regulators involved in mitochondrial biogenesis were assessed along SVZ-derived NSC differentiation. Unexpectedly, no significant alterations were found in PGC1α and TFAM protein levels. Simultaneously, overall AMPK and mTOR protein levels were also unchanged. Overall, these results suggest that mitochondrial biogenesis is not a major player in SVZ-derived NSC differentiation.

Importantly, our data revealed that mitochondrial dynamics is a key player in NSC maintenance and differentiation. In embryonic human and mouse NSCs, enhanced mitochondrial fusion promotes self-renewal, whereas increase of mitochondrial network fragmentation favors neuronal differentiation and maturation (Khacho et al, 2016; Iwata et al, 2020). Moreover, unique metabolic programs, namely, alterations in mitochondrial OXPHOS, lipid metabolism, reactive oxygen species (ROS) signaling, redox state, and glutaminolysis mark the transition between cellular stages along both embryonic and adult NSC lineages (Cunningham et al, 2007; Prozorovski et al, 2008; Ahlqvist et al, 2012; Stoll et al, 2015; Khacho et al, 2016; Namba et al, 2020; Adusumilli et al, 2021; Ramosaj et al, 2021; Wani et al, 2022). Noteworthily, mitochondrial dynamics was shown to regulate embryonic mouse NSC fate by fine-tuned regulation of ROS (Khacho et al, 2016). In addition, ROS was shown to influence the choice between neuronal and astroglial differentiation (Prozorovski et al, 2008). Therefore, one could postulate that these

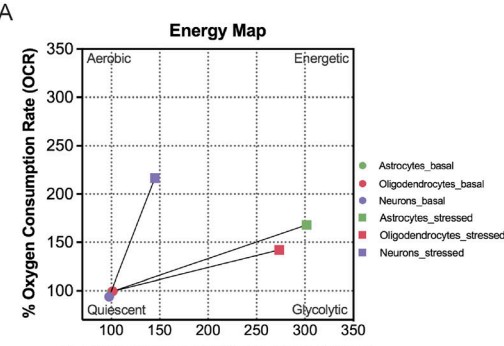

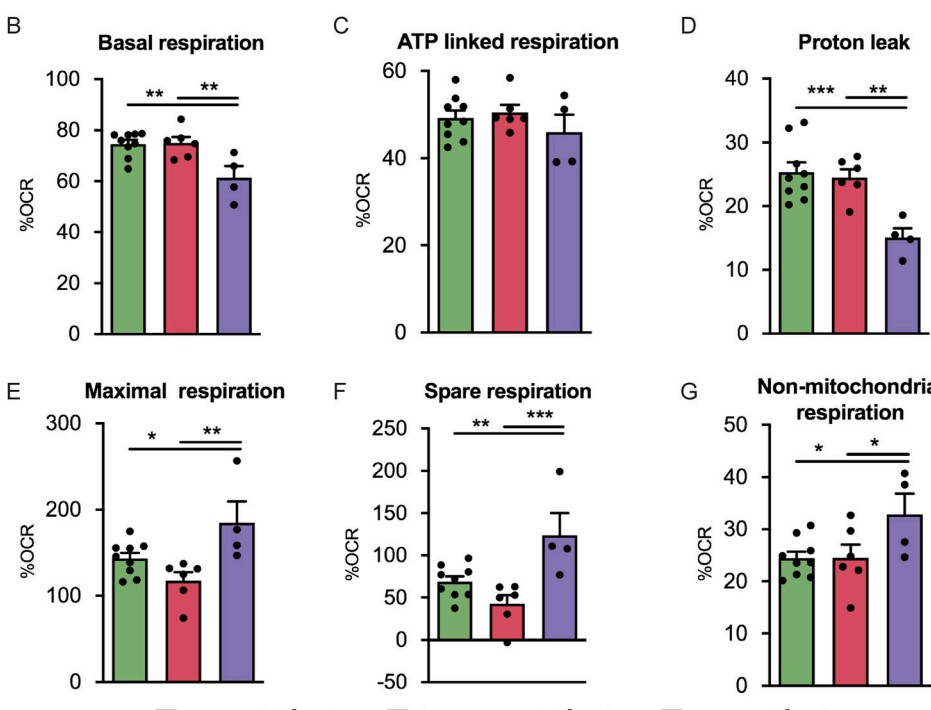

**Figure 6. Neurons are the most energetically flexible cells differentiated from the subventricular zone–derived neural stem cells, whereas oligodendrocytes are the least flexible. (A, B, C, D, E, F, G)** Energy map, (B) quantification of basal respiration, (C) ATP-linked respiration, (D) proton leak upon response to oligomycin, (E) maximal and (F) spare respiration upon response to FCCP, and (G) non-mitochondrial respiration upon response to rotenone and antimycin A. Data are presented as mean ± SEM (*P < 0.05 and **P < 0.01, one-way ANOVA followed by Fisher's least significant difference test).

mitochondrial processes could mediate the NSCs fate. Indeed, we have demonstrated that alterations in Mfn1, Mfn2, and DRP1 are a consequence of SVZ-derived NSC differentiation and are independent of mitochondrial mass alterations. Our results are consistent with studies demonstrating that Mfn1/2 deletion impairs neurogenesis, leading to cognitive deficits in adult mice (Fang et al, 2016; Khacho et al, 2016). Surprisingly, in our cell culture system, DRP1 protein levels decreased along NSC differentiation, independently of mitochondrial mass changes. These data are not in accordance with other studies both in mice and in humans, showing that mitochondrial fragmentation promotes neuronal differentiation over NSC self-renewal (Khacho et al, 2016; Iwata et al, 2020). Noteworthily, these studies were performed in embryonic NSCs, whereas for our studies, we use postnatal NSCs. Furthermore, it has been reported that mitochondrial morphology presents a different profile throughout embryonic and adult mouse neuronal differentiation (Khacho et al, 2016; Beckervordersandforth et al, 2017).

In this study, we also characterized the role of mitochondrial dynamics in postnatal NSC fate decision by evaluating the mitochondrial network in each cell type. We show that both NSCs and cell lineage differentiated from NSCs displayed distinct mitochondrial morphology, further suggesting that alterations in the mitochondrial dynamics impact differently on NSC commitment into the three lineages. Interestingly, in MBP+ cells, mitochondrial number, individuals, and networks decreased with a subsequent reduction in mitochondrial area with oligodendroglial maturation. Despite the limited findings of the mitochondrial phenotype in oligodendroglial cells, mitochondrial function reveals to be required for proper oligodendrocyte differentiation and myelination (Schoenfeld et al, 2010; Rinholm et al, 2016). In fact, mitochondrial morphology genes were induced as a consequence of oligodendroglial differentiation (Schoenfeld et al, 2010). This suggests that differentiated oligodendrocytes required altered mitochondrial morphology. Curiously, the number of mitochondria was higher in neurons when compared with more mature oligodendrocytes,

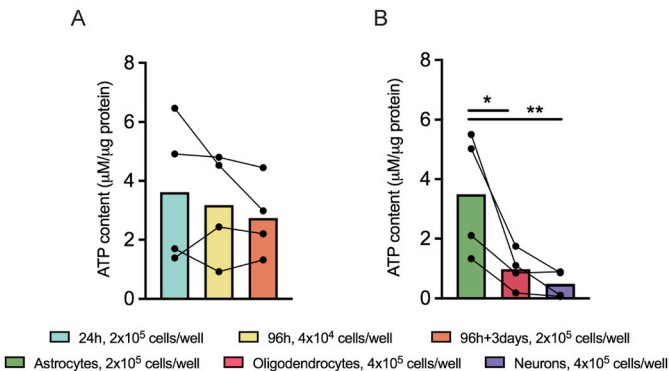

**Figure 7. Astrocytes have a significantly higher ATP content when compared with oligodendrocytes and neurons.**
**(A, B)** Quantification of the ATP content in (A) 24, 96, and 96 h + 3 d controls and (B) sorted cells. Data are presented as mean ± SEM (one-way ANOVA followed by Tukey's multiple comparison test for the comparison of the controls and one-way ANOVA followed by Fisher's least significant difference test for the comparison of the sorted cells).

which is in accordance with previous studies (Chang et al, 2006; MacAskill & Kittler, 2010). Although studies have shown that mitochondrial length is higher in astrocytes and neurons (neuronal dendrites and axons) than in oligodendrocytes (Rintoul et al, 2003; Chang et al, 2006; Jackson et al, 2014), in our work, the overall mitochondrial length was highly similar among the three lineage differentiated cells. Importantly, in contrast with the mentioned studies, in our cultures, the astrocytes, oligodendrocytes, and neurons are obtained from the differentiation of postnatal NSCs.

Because mitochondrial dynamics and morphology are at a certain extent linked to mitochondrial bioenergetics, the dissimilarities observed in the different fates derived from NSCs could be because of differences in the bioenergetic profile. Thus, a detailed bioenergetic profile was assessed during NSC differentiation and within each lineage specific cells. Interestingly, our data show that in the later stages of NSC differentiation, the cells require more ATP production. Importantly, a wide range of studies have demonstrated that differentiation into neurons, both in the embryonic and adult phase, is accompanied by a metabolic switch, shifting from glycolysis into OXPHOS (O'Brien et al, 2015; Khacho et al, 2016; Beckervordersandforth et al, 2017). In fact, the energy map of the sorted cells revealed that neurons present a more aerobic and energetic profile when exposed to stress conditions comparing with astrocytes and oligodendrocytes. These findings are in accordance with the significantly higher spare respiration and maximal respiration capacities of the neurons compared with both glial cells. This suggests that neurons have a higher bioenergetic flexibility, a trade that is required for handling constant energetically demanding processes, such as synaptic transmission and neuronal plasticity (Li & Sheng, 2022). On the other hand, oligodendrocytes are the least energetically flexible cells out of the differentiated cells. This hypothesis is corroborated by a study that demonstrates that human oligodendrocytes under conditions of increased stress, such as a low-glucose condition, revealed an overall decrease in OCR because of mainly a reduction in mitochondrial ATP-linked OCR (Rone et al, 2016). Intriguingly, and contrary to OPCs, post-myelinating

oligodendrocytes shift into primarily glycolytic metabolism (Fünfschilling et al, 2012), depending more on fatty acid synthesis (Dimas et al, 2019). These findings add another layer of complexity to our system, highlighting the heterogeneity of the energetic flexibility profile of cells within the same lineage. These dissimilarities between OPCs and oligodendrocytes could explain our results attained for morphological differences between the mitochondria from oligodendrocytes with different degrees of maturation. Interestingly, although no differences were observed in the ATP-linked respiration among the sorted cells, astrocytes present a significantly higher ATP content compared with the other cell types. This might suggest that in astrocytes, the ATP is also being produced by other pathways. In fact, evidences have shown that these glial cells have a predominantly glycolytic profile (Hamberger & Hydén, 1963; Bélanger et al, 2011), whereas the precise extent of OxPhos activity remains poorly understood. However, astrocytes that lack mitochondrial respiration were shown to survive as glycolytic cells (Supplie et al, 2017). These studies highlight the flexibility of the astrocytes to adapt their cellular energy state according to energy demand (Loaiza et al, 2003; Hertz et al, 2007). To further dissect the contributions of other pathways for the ATP production, complementary experiments to assess how reliant these neural cells are toward glycolysis, and more specifically for using pyruvate as a main fuel source, should be performed.

This work will certainly provide valuable new insights into molecular pathways that are unique for NSC fate and potentially unveil the importance of mitochondrial function in these processes. Notably, these data are groundwork to drive neural fate decision by modulating mitochondrial intrinsic properties and further explore the applicability of putative engineered NSCs in neural loss–associated disorders, such as Parkinson's disease and multiple sclerosis.

## Materials and Methods

### Animals

Mice were obtained from the iMM|JLA Rodent Facility (Lisbon, Portugal), where they were housed in a temperature-controlled room at 20–24°C. All the procedures were approved by the Portuguese National Authority for Animal Health (DGAV) and by the institute's animals' well-being office (ORBEA-iMM). This study was carried out in compliance with the ARRIVE guidelines (Percie du Sert et al, 2020). All experiments were performed in accordance with the European Community (86/609/EEC; 2010/63/EU; 2012/707/EU) and Portuguese (DL113/2013) legislation for the protection of animals used for scientific purposes.

### In vitro cultures

SVZ NSCs were obtained from early postnatal (P1-3) C57BL/6 mice. This model is appropriate to mimic the physiological context of the postnatal NSC differentiation. Neurosphere culture was performed as previously described (Soares et al, 2020). SVZ were seeded at a

**Table 1.   List of materials.**

| Reagent or resource | Manufacturer | Identifier |
| --- | --- | --- |
| Antibodies—immunocytochemistry | | |
| Rabbit polyclonal αtau | Synaptic Systems | Cat# 314003 |
| Rabbit polyclonal Chondroitin Sulphate Proteoglycan NG2 | Millipore | Cat# AB5320 |
| Rat monoclonal CD140a (PDGFRα) | BD Biosciences | Cat# 558774 |
| Rabbit polyclonal Doublecortin (DCX) | Abcam | Cat# ab18723 |
| Rabbit polyclonal Glial Fibrillary Acidic Protein (GFAP) | Sigma-Aldrich | Cat# G9269-.2 Ml |
| Mouse monoclonal HSP60 | BD Biosciences | Cat# 611562 |
| Rabbit monoclonal Myelin Basic Protein (MBP) | Cell Signaling Technology | Cat# 78896S |
| Rabbit polyclonal SOX2 | Abcam | Cat# ab97959 |
| Alexa Fluor 568 donkey anti-mouse | Thermo Fisher Scientific | Cat# A10037 |
| Alexa Fluor 488 donkey anti-rabbit | Thermo Fisher Scientific | Cat# A21206 |
| Alexa Fluor 488 donkey anti-rat | Thermo Fisher Scientific | Cat# A21208 |
| Antibodies—Western blot | | |
| Rabbit monoclonal AMPKα | Cell Signaling Technology | Cat# 5831 |
| Rabbit monoclonal p-AMPKα (Thr 172) | Cell Signaling Technology | Cat# 2535 |
| Rabbit monoclonal AMPK β1/2 | Cell Signaling Technology | Cat# 4150 |
| Rabbit p-AMPK β1 (Ser 108) | Cell Signaling Technology | Cat# 4181 |
| Mouse monoclonal PGC-1α | Millipore | Cat# ST1202 |
| Mouse monoclonal TFAM | Santa Cruz | Cat# sc-166965 |
| Rabbit monoclonal mTOR | Cell Signaling Technology | Cat# 2983 |
| Mouse monoclonal Mfn1 | Abnova | Cat# H00055669-M04 |
| Mouse monoclonal Mfn2 | Abnova | Cat# H00009927-M03 |
| Mouse monoclonal OPA1 | BD Biosciences | Cat# 612607 |
| Mouse monoclonal DLP1 | BD Biosciences | Cat# 611112 |
| Mouse monoclonal HSP60 | BD Biosciences | Cat# GC231-4H8 |
| Mouse monoclonal GAPDH | Thermo Fisher Scientific | Cat# AM4300 |
| Goat anti-Mouse HRP | Bio-Rad | Cat# 1706516 |
| Goat anti-Rabbit HRP | Bio-Rad | Cat# 1706515 |
| Magnetic beads | | |
| Anti-ACSA-2 MicroBead Kit | Miltenyi Biotec | 130-097-678 |
| Anti-O4 MicroBeads | Miltenyi Biotec | 130-096-670 |
| Neuron Isolation Kit | Miltenyi Biotec | 130-115-389 |
| Chemicals | | |
| PFA | Thermo Fisher Scientific | Cat# 15710 |
| B-27 Supplement (50X), serum-free | Thermo Fisher Scientific | Cat# 17504044 |
| Penicillin-Streptomycin | Thermo Fisher Scientific | Cat# 15140122 |
| EGF | Thermo Fisher Scientific | Cat# 53003018 |
| FGF | Thermo Fisher Scientific | Cat# 13256029 |
| Poly-D-Lysine 100 mg | Sigma-Aldrich | Cat# P7886 |
| Laminin | Sigma-Aldrich | Cat# L2020 |
| DAPI | Sigma-Aldrich | Cat# D9564 |
| Oligomycin | Sigma-Aldrich | Cat# O4876 |

**Table 1.  Continued**

| Reagent or resource | Manufacturer | Identifier |
|---|---|---|
| FCCP (Carbonyl cyanide-4- (trifluoromethoxy) phenylhydrazone) | Sigma-Aldrich | Cat# C2920 |
| Rotenone | Sigma-Aldrich | Cat# R8875 |
| Antimycin A | Sigma-Aldrich | Cat# A8674 |
| Critical commercial assays | | |
| NeuroCult Chemical Dissociation Kit (Mouse) | Stem Cell | Cat# 5707 |
| Luminescent ATP Detection Assay Kit | Abcam | Cat# ab113849 |
| Experimental models | | |
| C57BL/6 Mice | Charles River Laboratories | |
| Software and algorithms | | |
| Fiji | Max–Planck-Gesellschaft | http://fiji.sc |
| MiNA macro | Valente et al (2017) | |
| ZEN | | |
| GraphPad Prism 9.0 | GraphPad Software | https://www.graphpad.com/scientific-software/prism |
| Adobe Illustrator CC | Adobe | |

density of $2 \times 10^4$ cells/ml in serum-free medium (SFM) composed by DMEM/F12 + GlutaMAX-I supplemented with 100 U/ml penicillin and 100 $\mu$g/ml streptomycin, 1% B27, 10 ng/ml EGF, and 5 ng/ml bFGF (proliferation conditions) for 6–8 and 10–12 d, respectively. When most of the SVZ neurospheres have a diameter 150–200 $\mu$m, two passages were performed to obtain higher yields of NSCs, as described (Soares et al, 2020). SVZ neurospheres were plated (at a density of ~60 neurospheres per well) onto glass coverslips coated with 100 $\mu$g/ml PDL in 24-well plates. For immunoblotting analysis, SVZ neurospheres (at a density of ~480 neurospheres per well) were plated in coated six-well plates. The neurospheres were maintained in SFM devoid of growth factors (differentiation conditions). After 24 h, the medium was replaced with fresh SFM devoid of growth factors and at 4 and 10 d in vitro (DIV4 and 10), half of the medium was renewed. The SVZ-derived NSCs were allowed to develop for a maximum of 14 d (DIV14).

## Immunoblot

Western blot analysis was performed to assess levels of proteins involved in mitochondrial dynamics and biogenesis during SVZ-derived NSC differentiation. SVZ neurospheres at P2 were plated and allowed to develop for 0, 2, 4, and 7 d in differentiation conditions. Cells were lysed with the lysis buffer composed by 1 mM EGTA, 250 mM sucrose, 5 mM Tris–HCl, and 1% Triton X-100, pH 7.4, supplemented with protease and phosphatase inhibitors. Protein concentration was measured by the Bradford method accordingly to the manufacturer's specifications. 10–30 $\mu$m of proteins were separated by SDS–PAGE on 4–15% polyacrylamide gels (Bio-Rad) and electrophoretically transferred into a 0.2-$\mu$m nitrocellulose membranes. Membranes were blocked with 5% milk powder or 5% BSA in Tris-buffered saline with 0.1% Tween-20 for 1 h at RT. Incubations with the primary antibodies against proteins involved in mitochondrial dynamics were performed overnight at 4°C (Table 1).

Secondary antibodies conjugated with the horseradish peroxidase enzyme were used and detected by the ECL chemiluminescent luminol substrate (Amersham) and imaged on the Amersham 680 equipment. To determine the AMPK activity, a ratio between the activated and total forms was calculated. GAPDH was chosen as a loading control because no changes were observed during NSC differentiation, as supported by other reports (Agostini et al, 2016; Zheng et al, 2016; Isaksen et al, 2020). HSP60 was also used as a loading control to normalize for the mitochondrial mass. In the analysis, protein levels were normalized to DIV0. Protein levels of PGC1$\alpha$, TFAM, HSP60, Mfn1, Mfn2, OPA1, and DRP1, which are not represented as fold changes relative to DIV0, are shown in Figs S8 and S9.

## Morphometric analysis

SVZ neurospheres were plated at a density of 60 neurospheres per well onto coated glass coverslip in 24-well plates. To evaluate the mitochondrial network in the different cell types, cells at DIV0, 2, 4, 7, and 14 were fixed with 4% PFA in PBS. Afterward, they were permeabilized and blocked with 0.5% Triton X-100 and 3% BSA in PBS. Cells were then incubated with primary antibodies (anti-SOX2, anti-GFAP, anti-DCX, anti-$\alpha$tau, anti-NG2/anti-PDGFR$\alpha$, and anti-MBP to identify NSCs, astrocytes, immature neurons, mature neurons, OPCs, and oligodendrocytes, respectively) diluted in PBS with 0.1% Triton X-100 and 0.3% BSA (wt/vol), overnight at 4°C, and then with the corresponding secondary antibodies in PBS for 2 h at RT (Table 1). Nuclei were stained with 1 $\mu$g/ml DAPI in PBS, followed by mounting with Mowiol fluorescent medium. Importantly, in addition to the differentiation process, in oligodendrocytes, the analysis was also performed accordingly to the maturation stage because of the clearly observed differences in the mitochondrial structure between less and more mature oligodendrocytes. The distinction between the two types of maturation stages was based on the

complexity of the oligodendrocyte branches. Less mature oligo-dendrocytes presented poorly branched processes, whereas more mature oligodendrocytes presented complex branched processes. Mitochondrial network was identified through HSP60 staining. Importantly, the choice of the DIVs was based on the higher abundance of each cell type in culture. The fluorescence images were photographed on a ZEISS Cell Observer Spinning Disk confocal equipped with the ZEN software, using a 63x objective.

The neural cells were manually selected and then the morphometric analysis was performed on the mitochondria channel. For that, the Mitochondrial Network Analysis (MiNA) ImageJ macro (Valente et al, 2017) that is based on the existing ImageJ plug-in Skeleton was used to evaluate the mitochondria morphology. To obtain a sharper image with high contrast and minimal noise, the "unsharp mask," CLAHE, and median filters were applied together with the kernel convolution. In differentiated cells, the values obtained were normalized to the cytoplasmic area. A minimum of 40 cells were acquired per condition.

## Magnetic-activated cell sorting of differentiated cells

SVZ tertiary neurospheres were collected and dissociated using the NeuroCult Chemical Dissociation Kit, as previously described. Cells were then plated in coated dishes with 100 $\mu$g/ml PDL, in SFM supplemented with 5 ng/ml EGF and 2.5 ng/ml FGF-2 (low EGF/FGF-2) at a density of $1 \times 10^5$ cells/cm$^2$. At 96 h of plating, differentiated cells were sorted by magnetic-activated cell sorting. For that, anti-ACSA-2 MicroBeads were used to sort astrocytes followed by incubation with anti-O4 MicroBeads to obtain oligodendrocytes. Finally, neurons were isolated from the ACSA-2$^-$/O4$^-$ fraction by depletion of the magnetically labeled cells using the Neuron Isolation Kit. Then, ATP assay and respiratory assays were performed in the sorted cells 96 h + 3 d (day 7) and 96 h + 4 d (day 8), respectively. In addition, cells plated only 24 h in low EGF/FGF-2 medium were used as a control population abundant in NSCs. Moreover, cells plated for 96 h in low EGF/FGF-2 medium was also used as a control condition representing the heterogenous population presented before the magnetic separation. In both controls, cells were plated for ATP and respiratory assays. Moreover, to mimic the technical procedures to which the sorted cells were subjected, after the dissociation of tertiary neurospheres, single cells were also plated for 96 h + 3 d (day 7) in low EGF/FGF-2 medium to perform ATP assays and for 96 h + 4 d (day 8) to do respiratory assays.

## Respiratory assays

Mitochondrial respiration was evaluated by measuring the OCRs performing the Mito Stress protocol. This experiment was performed in the Seahorse Extracellular Flux (XF) 24 Analyzer (Seahorse Bioscience Agilent). The 24, 96, and 96 h + 4 d CTRs were plated onto 24-well Seahorse plates at the densities of $5 \times 10^4$, $1 \times 10^4$, and $5 \times 10^4$ cells/well, respectively. Regarding the sorted cells, astrocytes, oligodendrocytes, and neurons were seeded at the densities of $1 \times 10^5$, $1 \times 10^5$, and $5 \times 10^5$ cells/well, respectively. The low EGF/FGF-2 medium was replaced by the XF base medium supplemented with 10 mM glucose, 2 mM L-glutamine, and 1 mM

sodium pyruvate, pH 7.4. After measurements of resting respiration, cells were treated sequentially with oligomycin (2.5 $\mu$M for controls and 1.5 $\mu$M for sorted cells) to measure the nonphosphorylating OCR, two injections of FCCP (first injection: 2.5 $\mu$M for the 24-h control and 2 $\mu$M for the remaining conditions; second injection: 0.5 $\mu$M for all the conditions) to get the maximal OCR, and antimycin A and rotenone (1 $\mu$M) to measure the extramitochondrial OCR. Each measurement was taken over a 3-min interval followed by 3 min of mixing and 2 min of incubation. Three measurements were taken for the resting OCR: after oligomycin treatment, after FCCP, and after antimycin A/rotenone treatment. OCR levels were normalized to protein levels. Protein concentration of the samples was determined using the Pierce BCA Protein Assay Kit as described by the manufacturer. Importantly, the %OCR was calculated by normalizing the raw OCR values (normalized only by protein) to the first measured point, and then the obtained values were multiplied by 100 to express them as a percentage. This normalization is crucial to eliminate the inherent basal respiration variability within each replicate.

## ATP content determination

To determine ATP content, a luciferase-based luminescent ATP determination assay was used according to the manufacturer's protocol (Abcam). All ATP content levels were normalized to protein levels.

## Statistical analysis

Data are represented as mean ± SEM or median ± interquartile range. Graphical illustrations and significance were obtained with GraphPad Prism 9 (GraphPad). Significance was calculated as described in each figure legend. Values of $P < 0.05$ were considered to represent statistical significance.

# Supplementary Information

# Acknowledgements

This work was supported by the European Molecular Biology Organization (EMBO), IG#3309; Fundação para a Ciência e Tecnologia (FCT) (PTDC/MED-NEU/7976/2020); International Society for Neurochemistry (ISN) Career Development Grant; and International Brain Research Organization (IBRO) Early Career Award. R Soares (PD/BD/128280/2017, COVID/BD/151619/2021, and IMM/BI/8-2021) and DM Lourenço (PD/BD/141784/2018 and COVID/BD/152658/2022) were in receipt of a fellowship from FCT. VA Morais is supported by FCT (IF/01693/2014; IMM/CT/27-2020). This project has received funding from H2020-WIDESPREAD-05-2020-Twinning (EpiEpinet) under grant agreement No 952455. We thank members of the VA Morais and AM Sebastião Labs for fruitful discussions. We would like to thank the BioImaging Facility, with a special thanks to José Rino, António Temudo, and Ana Nascimento and the Rodent Facility of Instituto de Medicina Molecular João Lobo Antunes for their technical support, and we also acknowledge the funding PPBI-POCI-01-0145-FEDER-022122.

## Author Contributions

R Soares: conceptualization, formal analysis, investigation, methodology, and writing—original draft, review, and editing.
DM Lourenço: methodology.
IF Mota: methodology.
AM Sebastião: funding acquisition, investigation, and writing—review and editing.
S Xapelli: conceptualization, formal analysis, supervision, funding acquisition, investigation, and writing—review and editing.
VA Morais: conceptualization, formal analysis, supervision, funding acquisition, investigation, and writing—review and editing.

## Conflict of Interest Statement

The authors declare that they have no conflict of interest.

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
