## [Reviewer comments · Life Science Alliance]

Life Science Alliance

Lineage-specific changes in mitochondrial properties during neural stem cell differentiation

Rita Soares, Diogo Lourenço, Isa Mota, Ana Sebastião, Sara Xapelli, and Vanessa Morais

DOI: <https://doi.org/10.26508/lsa.202302473>

Corresponding author(s): Vanessa Morais, Instituto de Medicina Molecular, Lisboa and Sara Xapelli, Faculdade de Medicina Universidade de Lisboa

Review Timeline:

Submission Date:	2023-11-07
Editorial Decision:	2023-12-27
Revision Received:	2024-02-26
Editorial Decision:	2024-03-25
Revision Received:	2024-04-06
Accepted:	2024-04-09

Transaction Report:

December 27, 2023

Re: Life Science Alliance manuscript #LSA-2023-02473

Prof. Vanessa A. Morais
Instituto de Medicina Molecular, Lisboa
Avenida professor Egas Moniz
Lisboa 1649-028
Portugal

Dear Dr. Morais,

Thank you for submitting your manuscript entitled "Lineage-specific changes in mitochondrial properties during neural stem cell differentiation" to Life Science Alliance. The manuscript was assessed by expert reviewers, whose comments are appended to this letter. We invite you to submit a revised manuscript addressing the Reviewer comments.

Thank you for this interesting contribution to Life Science Alliance. We are looking forward to receiving your revised manuscript.

Sincerely,

B. MANUSCRIPT ORGANIZATION AND FORMATTING:

Reviewer #1 (Comments to the Authors (Required)):

In this interesting study, Soares et al study various aspects of mitochondrial biology during differentiation of mouse SVZ-derived neurospheres. They first quantify levels of mitochondrial biogenesis-related proteins, which are mostly not changing, and then perform a thorough investigation of mitochondrial morphology at various stages of differentiation and in different cell-types. Finally, they perform Seahorse experiments in the same stages and cell-types. Interestingly, many of their measurements do not show significant changes between different conditions or stages, and I feel this is understated throughout the manuscript, but equally interesting.

This is an interesting manuscript, with relatively limited scope, but potentially interesting and novel findings, providing significant insight into how mitochondrial metabolism may change during neurogenesis. The manuscript is well written, although the conclusions in the introduction, results and discussion are sometimes overstated and lack specifics. The figures are clear, and the experiments seem overall well conducted.

I am supportive of publication in principle, but have several major and minor concerns, listed below, that should be addressed first.

- Line 42: the introduction does not refer to differences between mouse and human adult neurogenesis, and some of the recent controversy around this topic (eg PMID: 29513649). Alternatively, it would be good to specify clearly which model systems the authors refer to. In addition, the authors should specify when they refer to work primarily performed in embryogenesis (eg Iwata et al, 2020) or adult neurogenesis. It might be worth referring to other model systems where mitochondrial metabolism/morphology has been studied in NSCs, for example Drosophila (eg PMID: 25126791, PMID: 31513013, PMID: 35157701).
- Results section 1: The authors should briefly explain the model system they use, explain that this is done in neurospheres, and how differentiation is done, to understand what the cell-type composition is at the various DIV.
- Fig. 1B: normalisation should be done to average of DIV0, to demonstrate baseline variability as well.
- Line 133: SVZ cells are mentioned here. Are these the same neurospheres as used in Fig1/2?
- Fig3: If the graphs show normalisation to HSP60, the example blots should ideally show HSP60 in each blot too, instead of GAPDH (which should be shown in S2, next to the graphs there).
- Fig4: quantifications in the respective lineages should be accompanied by representative images, also illustrating the differences between different morphologies. If cells were manually selected (see Methods), what were the criteria to include/exclude specific cells? If at least 40 cells were quantified, how many different replicates (cultures) were quantified, and what do the dots represent on the graphs? In addition, the discrete nature of the data (eg striking in Fig4C Astrocytes DIV4) is not entirely clear to me when normalised to cytoplasmic area, which should be a continuous variable?
- A relevant valuable that is missing from Fig4 is the average length (and length distribution) of mitochondria in each cell type and time point, as a measure for mitochondrial fragmentation, and fission/fusion. In particular upon differentiation in immature neurons, where this measure is relevant in embryonic neurogenesis (PMID: 32792401), and to test whether similar dynamic changes occur during adult neurogenesis (eg as discussed in PMID: 35714855 or PMID: 36445292).
- Line 197: correlation is described, which is not an indication of causality, so it would be good to discuss this differently.
- Figure 5 and lines 212-214: I am not sure the main conclusion of these results is that mitochondria rely more on mitochondria, if the only difference observed among many tests, is a subtle, non-progressive, increase in ATP-linked respiration? This conclusion should be revised or toned down.
- Figure 5/6: it is not entirely clear how %OCR is calculated, in particular if for example basal respiration (5B and 6B) is not 100%?

Minor comments:

- In the abstract, the difference between 'number' (line 36) and 'area' (line 37) is not clear initially.
- Line 71: also mtDNA replication.
- Line 104: PGC1a and TFAM do not interact in the same way as other transcription factors do, since TFAM is mitochondrial.
- Line111: previous
- Line203: control control
- Line 274: Dohla et al, 2022 does not describe NSCs, if I'm not mistaken.

Reviewer #2 (Comments to the Authors (Required)):

In their manuscript "Lineage-specific changes in mitochondrial properties during neural stem cell differentiation" Soares and colleagues used neural stem cells (NSCs) isolated from the subventricular zone (SVZ) of the adult mouse to characterize mitochondrial dynamics and bioenergetic demands of proliferating and differentiating NSCs cultured in vitro. They show that mitochondrial dynamics show cell-type-specific (e.g., neuronal vs. astroglial vs. oligodendroglial) changes upon differentiation. Furthermore, they find that astroglial cells have a higher ATP content compared to neurons and oligodendrocytes. The experimental approach is straightforward and the data are convincing. However, the conceptual advance provided is limited: as (openly) discussed by the authors there are substantial data on the role of mitochondrial metabolism for both embryonic and adult neurogenesis (e.g., by Vanderhaeghen, Lie, Knobloch, Slack groups...). The data shown here add to this previous work - and will be of interest to specialized readers in the field. However, the study suffers from the fact that it is based exclusively on in vitro data. If the conclusions drawn here hold true for the in vivo situation remains unclear (the authors write in their introduction line 88: "...Therefore, the aim of this work was to assess how mitochondrial biogenesis and dynamics change along postnatal NSC in the SVZ niche....". Unfortunately, the whole study is based on cells outside of their niche). Indeed, culture conditions will have a major impact on metabolism, e.g., by high glucose conditions in the medium. Thus, an approach exclusively based on cultured NSCs may be suitable if truly novel mechanistic insights will be discovered. However, in its current form the study is certainly sound (and as such convincing). But as outlined above: the advance provided is rather limited.

**Life Science Alliance manuscript
LSA-2023-02473**

Full Title:

Lineage-specific changes in mitochondrial properties during neural stem cell differentiation

Authors: Rita Soares, Diogo M. Lourenço, Isa F. Mota, Ana M. Sebastião, Sara Xapelli, Vanessa A. Morais

COMMENTARIES TO EDITOR AND REVIEWERS

Dear Editor,

We would like to thank the editor and the referees for the thorough review of our submitted manuscript "Lineage-specific changes in mitochondrial properties during neural stem cell differentiation".

Regarding their suggestions, we believe that they have improved the paper and we are convinced that, in its present form, the manuscript addresses the main criticisms raised by the reviewers.

We are now submitting the revised version of the manuscript with track changes.

We hope that you will find that this new version, together with the specific point-by-point replies made below to the editor and referees, are adequate for the publication of this study in Life Science Alliance.

Best regards,

Vanessa A. Morais

Response to Reviewer 1:

In this interesting study, Soares et al study various aspects of mitochondrial biology during differentiation of mouse SVZ-derived neurospheres. They first quantify levels of mitochondrial biogenesis-related proteins, which are mostly not changing, and then perform a thorough investigation of mitochondrial morphology at various stages of differentiation and in different cell-types. Finally, they perform Seahorse experiments in the same stages and cell-types. Interestingly, many of their measurements do not show significant changes between different conditions or stages, and I feel this is understated throughout the manuscript, but equally interesting.

This is an interesting manuscript, with relatively limited scope, but potentially interesting and novel findings, providing significant insight into how mitochondrial metabolism may change during neurogenesis. The manuscript is well written, although the conclusions in the introduction, results and discussion are sometimes overstated and lack specifics. The figures are clear, and the experiments seem overall well conducted.

I am supportive of publication in principle, but have several major and minor concerns, listed below, that should be addressed first.

Major comments:

Line 42: the introduction does not refer to differences between mouse and human adult neurogenesis, and some of the recent controversy around this topic (eg PMID: 29513649). Alternatively, it would be good to specify clearly which model systems the authors refer to. In addition, the authors should specify when they refer to work primarily performed in embryogenesis (eg Iwata et al, 2020) or adult neurogenesis. It might be worth referring to other model systems where mitochondrial metabolism/morphology has been studied in NSCs, for example Drosophila (eg PMID: 25126791, PMID: 31513013, PMID: 35157701).

We thank the Reviewer for their comment concerning the reference of the used models and the type of NSCs (development vs adult), used in the cited work. We have reviewed all the sections of the manuscript and we have now included in the revised manuscript the missing information in the Introduction (lines 58, 74, 84, 89, 95, 97 and 98) and Discussion (lines 305, 313, 314, 320, 323 and 354) sections. Additionally, we have also specified the model used in our study, i.e. mouse and the type NSCs, i.e. postnatal, in the Abstract (lines 30 and 41), Introduction (line 102) and Discussion (lines 297, 328 and 348).

Regarding other model systems where mitochondrial metabolism/morphology has been studied in NSCs, we have added this information in the Introduction (lines 90-93 and 97-98).

The recent controversy surrounding adult neurogenesis in humans has been incorporated in the Introduction section (lines 44-50). We included possible explanations (such as used samples and methodology) to address the variations observed in the studies.

Results section 1: The authors should briefly explain the model system they use, explain that this is done in neurospheres, and how differentiation is done, to understand what the cell-type composition is at the various DIV.

We fully agree with the Reviewer's comment concerning the lack of explanation regarding the neurosphere assay model (NSA) and how differentiation process is induced. Therefore, we have added in the Results section 1 of the revised manuscript, information about the main advantages of the NSA model, the neurogenic region where NSCs were isolated, the purpose of neurosphere passage, the induction of differentiation, and finally, the cellular composition throughout differentiation (lines 114-128).

Fig. 1B: normalisation should be done to average of DIV0, to demonstrate baseline variability as well.

Concerning reviewers' question about the variability of the data at DIV0 (baseline), fold change for the values at DIV2, DIV4, and DIV7 was determined in each culture (N=1) in comparison to the values at DIV0. All the samples from the same N were loaded in the same Western Blot gel. Since per culture, we

only have one value per DIV, corresponding to the quantification of one band, averaging is not possible per N. On the other hand, in our opinion, normalization to the average of DIV0 is not the most appropriate by combining all the cultures because each blot has its own variability, including the exposure time used for the acquisition and the binding of both primary and secondary antibodies.

Line 133: SVZ cells are mentioned here. Are these the same neurospheres as used in Fig1/2?

The SVZ cells mentioned in line 163 of the revised manuscript (corresponding to line 133 of the first version of the manuscript) are the same neurospheres as those shown in Figures 1 and 2. Notably, all immunoblot experiments were conducted under the same conditions, meaning with tertiary neurospheres (neurospheres subjected to 2 passages). This information is detailed in lines 409-414 of the *In vitro* cultures section of the Materials and Methods.

Fig3: If the graphs show normalisation to HSP60, the example blots should ideally show HSP60 in each blot too, instead of GAPDH (which should be shown in S2, next to the graphs there).

Concerning the absence of the HSP60 example blots in Figure 3, we did not include them for all the proteins (Mfn1, Mfn2, OPA1 and DRP1) due to technical limitations that prevented us from obtaining HSP60 blots for each analysed protein. In fact, we evaluated eleven proteins (biogenesis- and fusion/fission-related proteins) and due to the limited cell yields we were unable to perform HSP60 normalization for all blots. Therefore, we normalized the levels of Mfn1, Mfn2, OPA1 and DRP1 proteins to GAPDH, as well as the levels of HSP60 protein to GAPDH. Finally, we calculated the ratio of the normalized fusion/fission-related protein levels to GAPDH with the normalized HSP60 protein levels to GAPDH. Worth mentioning is that no significant differences of HSP60 protein levels were observed throughout NSC differentiation (Fig3 A, B), thus, validating this approach for normalization.

Fig4: quantifications in the respective lineages should be accompanied by representative images, also illustrating the differences between different morphologies. If cells were manually selected (see Methods), what were the criteria to include/exclude specific cells? If at least 40 cells were quantified, how many different replicates (cultures) were quantified, and what do the dots represent on the graphs? In addition, the discrete nature of the data (eg striking in Fig4C Astrocytes DIV4) is not entirely clear to me when normalised to cytoplasmic area, which should be a continuous variable?

The Reviewer has raised important questions regarding the quantification of the mitochondrial morphology in cells from different lineages.

Regarding the representative images of the mitochondrial morphology results presented in Figure 4, in the first submission we included a representative image

of the mitochondrial structures in each cell lineage in FigS4, and in NSCs in FigS3. We now include one image per DIV per cell type in two figures (FigS4 and S5 modified) (lines 858-865). For clarity, we have now included in the text, along with the Fig4, the respective representative image in FigS4 and FigS5 (lines 193, 202, 204, 205, 207, 213, 215, 217, 218, 219, 221, 222, 224, 225 and 226). Additionally, for reader convenience, we have included in FigS4 and FigS5 the name of each cell type to the respective cellular marker(s). With the inclusion of this additional figure, the previous FigS5 and FigS6 are now the FigS6 and FigS7, respectively. This alteration was updated in the manuscript (lines 246, 260, 870, 872, 875 and 877).

As NSCs in differentiative conditions give rise to various cell types from distinct lineages, assessing mitochondrial morphology in a specific cell type requires combining the mitochondrial marker HSP60 with a neural marker, such as SOX2 for NSCs (FigS3) and DCX for immature neurons (FigS4). Therefore, when we mention "The neural cells were manually selected" (in line 459 of the revised manuscript) it means that we manually defined a region of interest (ROI) that included only the positive cells for a specific neural marker, excluding the negative ones from the analysis. For example, in FigS3 we manually created a ROI with only SOX2-positive cells to evaluate the mitochondrial signal originating from the NSC population.

Concerning the number of replicates (cultures), we performed quantifications from 2 to 5 independent cultures. The dots, representing each "n", denote the cells that were analysed. To be clear, we have included this information in the legend of Fig4 (lines 776-778) and FigS3 (line 857).

Regarding reviewer's comment about the discrete nature of the data, this is due to the presence of a heterogenous population of overlapping cells, therefore, the analysis of the mitochondrial morphology and the cytoplasmic area had to be done in more than one cell at a time. Consequently, the average values of mitochondrial parameters and cytoplasmic area were assessed, and the result was recorded in agreement with the number of cells analysed. As an illustration, the data in myelinating oligodendrocytes (MBP+ cells) exhibit less discreteness compared to astrocytes, as oligodendrocytes tend to migrate more, and thus are not overlapped with other cells.

A relevant valuable that is missing from Fig4 is the average length (and length distribution) of mitochondria in each cell type and time point, as a measure for mitochondrial fragmentation, and fission/fusion. In particular upon differentiation in immature neurons, where this measure is relevant in embryonic neurogenesis (PMID: 32792401), and to test whether similar dynamic changes occur during adult neurogenesis (eg as discussed in PMID: 35714855 or PMID: 36445292).

We are in full agreement with reviewer's comment about the absence of calculations for the average length and length distribution of mitochondria in each cell type and time point. Although our analysis of mitochondrial length – "Rod & branch length" - does not account for puncta mitochondria (single pixel in the skeletonized image) representing fragmented mitochondria, we

measured other parameters (shown in Figure 4A). These parameters, when considered together, allow to draw conclusions about mitochondrial fragmentation and fusion/fission events including the number of individuals (such as large/round, rod, and punctate structures), the number of networks and the number of branches per network. To clarify, we have included this information in the Results section, presenting the mitochondrial morphology data (lines 198-199). As demonstrated in a previous article describing the MiNA macro (Valente et al, 2017), changes in MiNA-related morphological parameters indeed reflected a condition in which mitochondrial fragmentation was induced, such as exposure to the mitochondrial uncoupler FCCP. Indeed, the number of mitochondrial individuals increased, along with the increased number of networks. This later observation was related to the fragmentation of larger networks with more branches into many smaller networks. In addition to this, an expected decrease in the number of branches per network was also observed. Moreover, with FCCP exposure, the mitochondrial footprint was reduced, possibly indicating mitophagy of the fragmented mitochondria.

Line 197: correlation is described, which is not an indication of causality, so it would be good to discuss this differently.

Regarding the conclusion of the mitochondrial morphology data, we agree with the Reviewer's comment. Indeed, we have now revised the conclusion (lines 229 and 232).

Figure 5 and lines 212-214: I am not sure the main conclusion of these results is that mitochondria rely more on mitochondria, if the only difference observed among many tests, is a subtle, non-progressive, increase in ATP-linked respiration? This conclusion should be revised or toned down.

We agree with the Reviewers' concern regarding the main conclusion of Fig5. Consequently, we have revised the title of the figure to "NSC differentiation at the later stages requires higher levels of ATP production" (line 235), adjusted the conclusion to "Overall, these data suggest that throughout NSC differentiation, particularly at the later time-points, there was a more pronounced reduction in the % of OCR when mitochondrial ATP production was inhibited." (lines 250-252) and the Discussion to "Interestingly, our data show that in the later stages of NSC differentiation the cells require more ATP production." (lines 353 and 355).

Figure 5/6: it is not entirely clear how %OCR is calculated, in particular if for example basal respiration (5B and 6B) is not 100%?

The %OCR was calculated by normalizing the raw OCR values (normalized only by protein) to the first measured point, and then the obtained values were multiplied by 100 to express them as a percentage. This normalization is crucial to eliminate the inherent basal respiration variability within each replicate, namely across the three time-points of NSC differentiation (24h, 96h and 96h+4days) (Fig5), and among different cell types (astrocytes, oligodendrocytes and neurons)(Fig6). Hence, it eliminates the variability

between different cultures (Ns). For clarity, we have now included this information in the Materials and Methods section, specifically in the Respiratory Assays subsection (lines 503-508).

The Basal respiration values shown in Fig5B and 6B are not set at 100% because this parameter specifically refers to the basal respiration linked to mitochondria. Thus, the non-mitochondrial respiration, measured after Rotenone and Antimycin A injections, was subtracted from the total (100%). Importantly, to show the differences in basal respiration levels at steady-state conditions, we included the OCR profiles without normalization to the first point in the manuscript (FigS6 and FigS7). For instance, our study demonstrates significant differences in various respiratory assay parameters in neurons compared to the other differentiated cells through %OCR (Fig6), while concurrently showing, via the OCR profile, that the neuronal population exhibits lower basal respiration under steady-state conditions (FigS7).

In summary, having the OCR profile expressed as a percentage and as raw values (normalized only by protein) provides a more informative analysis, and indicates the overall OCR levels of different neural cells at basal levels.

Minor comments:

In the abstract, the difference between 'number' (line 36) and 'area' (line 37) is not clear initially.

Regarding the Abstract, the word “number” was changed to “number of branched and unbranched mitochondria” (lines 36 and 37) and the word “area” was replaced by “area occupied by mitochondrial structures” (lines 38).

Line 71: also mtDNA replication.

As suggested by the Reviewer, mtDNA replication was added to the following sentence (line 78):

*Mitochondrial biogenesis is the formation of de novo mitochondria from pre-existing ones, which requires mitochondrial DNA (mtDNA) **replication**, transcription and translation (Popov, 2020).*

Line 104: PGC1 α and TFAM do not interact in the same way as other transcription factors do, since TFAM is mitochondrial.

We agree with the Reviewers' comment about the interaction of PGC1 α with transcription factors other than TFAM, and to clarify this we have revised the sentence, briefly explaining the PGC1 α -TFAM interaction (lines 132-135).

Line111: previous

The word “previsous” was changed to “previous” (line 141).

Line203: control control

One of the words "control" was eliminated (line 240).

Line 274: Dohla et al, 2022 does not describe NSCs, if I'm not mistaken.

Regarding the reference to Dohla et al, we agree with the Reviewer's observation about the absence of NSC usage in this study. In fact, Dohla et al utilized epithelial stem-like cells and not NSCs. Therefore, we have removed this reference in the revised manuscript (lines 311-313).

Response to Reviewer 2:

In their manuscript "Lineage-specific changes in mitochondrial properties during neural stem cell differentiation" Soares and colleagues used neural stem cells (NSCs) isolated from the subventricular zone (SVZ) of the adult mouse to characterize mitochondrial dynamics and bioenergetic demands of proliferating and differentiating NSCs cultured in vitro. They show that mitochondrial dynamics show cell-type-specific (e.g., neuronal vs. astroglial vs. oligodendroglial) changes upon differentiation. Furthermore, they find that astroglial cells have a higher ATP content compared to neurons and oligodendrocytes.

The experimental approach is straightforward and the data are convincing. However, the conceptual advance provided is limited: as (openly) discussed by the authors there are substantial data on the role of mitochondrial metabolism for both embryonic and adult neurogenesis (e.g., by Vanderhaeghen, Lie, Knobloch, Slack groups...). The data shown here add to this previous work - and will be of interest to specialized readers in the field. However, the study suffers from the fact that it is based exclusively on in vitro data. If the conclusions drawn here hold true for the in vivo situation remains unclear (the authors write in their introduction line 88: "...Therefore, the aim of this work was to assess how mitochondrial biogenesis and dynamics change along postnatal NSC in the SVZ niche....". Unfortunately, the whole study is based on cells outside of their niche). Indeed, culture conditions will have a major impact on metabolism, e.g., by high glucose conditions in the medium. Thus, an approach exclusively based on cultured NSCs may be suitable if truly novel mechanistic insights will be discovered. However, in its current form the study is certainly sound (and as such convincing). But as outlined above: the advance provided is rather limited.

Although there are several publications regarding the role of mitochondrial metabolism for both embryonic and adult neurogenesis (Khacho et al, 2016; Beckervordersandforth et al, 2017; Iwata et al, 2020, 2023; Petrelli et al, 2023), in our humble opinion, our article provides new detailed and relevant information. We evaluated the role of mitochondrial metabolism, as well as mitochondrial fusion/fission events, mitochondrial morphology and mitochondrial biogenesis, throughout adult NSC differentiation. This evaluation encompasses not only

neurons (neurogenesis) but also other lineages, namely oligodendroglial and astroglial, simultaneously.

We are in full agreement with the Reviewer that this work would have had a more relevant impact with the inclusion of *in vivo* data. However, in our opinion, to further understand the involvement of mitochondrial properties in adult NSC fate decisions *in vivo*, it is essential to first comprehend these mitochondrial processes in each neural cell type and during adult NSC differentiation. Therefore, choosing an *in vitro* system that allows spontaneous differentiation of adult NSCs is crucial as a starting point.

Accordingly, we decided to use the NSA model in our work as the *in vitro* setup. It is important to note that this model is well-established in our laboratory and has been fully characterized by us (Soares et al, 2020, 2021). Indeed, over the last decades the NSA has been proven to be a powerful tool to study the NSC behavior *in vivo* mainly by mimicking the heterogeneity of the niche. Moreover, this model has been shown to be a valuable tool to evaluate putative regulators of NSC fate. By maintaining the cells in a serum-devoid medium, the environmental cues are solely provided by the surrounding environment (Jensen & Parmar, 2006; Galli, 2019).

In our work, we isolated the adult NSCs from the SVZ, one of the main neurogenic niches. Despite the SVZ cells being outside the niche, when isolated, they still retain their endogenous properties, notably the intrinsic tri-potential. In light of this issue and considering the Reviewer's comment about the sentence in line 88 of the Introduction of the original manuscript that mentions "...postnatal mouse NSC in the SVZ niche...", we have rephrased this sentence to "...postnatal SVZ-derived mouse NSC" (lines 102 and 103). Additionally, we have also replaced the words "SVZ niche" and "SVZ NSCs" with "SVZ-derived NSCs" (lines 138, 142, 153, 254, 272, 273, 788 and 789).

Finally, we understand the Reviewer's concern about the impact of the high glucose concentrations in the medium on mitochondrial metabolism. Nonetheless, the choice of DMEM/F-12+GlutaMAX medium was made based on the best conditions for the adult NSC maintenance and differentiation, as used by others researchers and published in relevant journals that included the NSA (Khacho et al, 2016; Bertolini et al, 2019; Lin et al, 2020).

Based on our findings, this work advanced our knowledge on the crosstalk between NSC differentiation and mitochondrial dynamics and bioenergetics, shedding light on the importance of modulating these mitochondrial processes in NSC research, which can nurture future studies using an *in vivo* approach, and ultimately could be used to develop new therapeutic strategies for brain repair.

References

- Beckervordersandforth R, Ebert B, Schöffner I, Moss J, Fiebig C, Shin J, Moore DL, Ghosh L, Trincherio MF, Stockburger C *et al* (2017) Role of Mitochondrial Metabolism in the Control of Early Lineage Progression and Aging Phenotypes in Adult Hippocampal Neurogenesis. *Neuron* 93: 560-573.e6. doi:10.1016/J.NEURON.2016.12.017.
- Bertolini JA, Favaro R, Zhu Y, Pagin M, Ngan CY, Wong CH, Tjong H, Vermunt MW, Martynoga B, Barone C *et al* (2019) Mapping the Global Chromatin Connectivity Network for Sox2 Function in Neural Stem Cell Maintenance. *Cell stem cell* 24: 462-476.e6. doi:10.1016/j.stem.2019.02.004.
- Galli R (2019) The Neurosphere Assay (NSA) Applied to Neural Stem Cells (NSCs) and Cancer Stem Cells (CSCs). (Humana Press, New York, NY), pp. 139–149.
- Iwata R, Casimir P, Erkol E, Boubakar L, Planque M, Gallego López IM, Ditkowska M, Gaspariunaite V, Beckers S, Remans D *et al* (2023) Mitochondria metabolism sets the species-specific tempo of neuronal development. *Science* 379. doi:10.1126/science.abn4705.
- Iwata R, Casimir P, Vanderhaeghen P (2020) Mitochondrial dynamics in postmitotic cells regulate neurogenesis. *Science* 369: 858–862. doi:10.1126/science.aba9760.
- Jensen JB, Parmar M (2006) Strengths and Limitations of the Neurosphere Culture System. *Molecular Neurobiology* 34: 153–162. doi:10.1385/MN:34:3:153.
- Khacho M, Clark A, Svoboda DS, Azzi J, MacLaurin JG, Meghaizel C, Sesaki H, Lagace DC, Germain M, Harper M-E *et al* (2016) Mitochondrial Dynamics Impacts Stem Cell Identity and Fate Decisions by Regulating a Nuclear Transcriptional Program. *Cell Stem Cell* 19: 232–247. doi:10.1016/j.stem.2016.04.015.
- Lin L, Zhang M, Stoilov P, Chen L, Correspondence SZ, Zheng S (2020) Developmental Attenuation of Neuronal Apoptosis by Neural-Specific Splicing of Bak1 Microexon II Developmental Attenuation of Neuronal Apoptosis by Neural-Specific Splicing of Bak1 Microexon. *Neuron* 107: 1180-1196.e8. doi:10.1016/j.neuron.2020.06.036.
- Petrelli F, Scandella V, Montessuit S, Zamboni N, Martinou J-C, Knobloch M (2023) Mitochondrial pyruvate metabolism regulates the activation of quiescent adult neural stem cells. *Science Advances* 9. doi:10.1126/sciadv.add5220.
- Soares R, Ribeiro F, Lourenço Di, Rodrigues R, Moreira J, Sebastião A, Morais V, Xapelli S (2021) The neurosphere assay: An effective in vitro technique to study neural stem cells. *Neural Regeneration Research* 16: 2229–2231. doi:10.4103/1673-5374.310678.
- Soares R, Ribeiro FF, Lourenço DM, Rodrigues RS, Moreira JB, Sebastião AM, Morais VA, Xapelli S (2020) Isolation and Expansion of Neurospheres from Postnatal (P1-3) Mouse Neurogenic Niches. *Journal of Visualized Experiments*: e60822. doi:10.3791/60822.
- Valente AJ, Maddalena LA, Robb EL, Moradi F, Stuart JA (2017) A simple ImageJ macro tool for analyzing mitochondrial network morphology in mammalian cell culture. *Acta Histochemica* 119: 315–326. doi:10.1016/j.acthis.2017.03.001.

March 25, 2024

RE: Life Science Alliance Manuscript #LSA-2023-02473R

Prof. Vanessa A. Morais
Instituto de Medicina Molecular, Lisboa
Avenida professor Egas Moniz
Lisboa 1649-028
Portugal

Dear Dr. Morais,

Thank you for submitting your revised manuscript entitled "Lineage-specific changes in mitochondrial properties during neural stem cell differentiation". We would be happy to publish your paper in Life Science Alliance pending final revisions necessary to meet our formatting guidelines.

- please address Reviewer 1's remaining comments
- please be sure that the authorship listing and order is correct
- please upload your main and supplementary figures as single files;
- manuscript file should be provided without figures, only with the figure legends placed after the reference section
- all figures should be uploaded individually without captions
- please add ORCID ID for the secondary corresponding author -- they should have received instructions on how to do so
- please add the Twitter handle of your host institute/organization as well as your own or/and one of the authors in our system
- please upload a clean version of the manuscript file without the track changes
- please add callouts for Figures S1A-B; S6A-B and SA-B to your main manuscript text

Figure Checks:

-In Figure 3A, the GAPDH blots used for Mfn1 and DRP1 appear to be the same. Please correct, or provide an explanation of why this would be possible.

A. FINAL FILES:

B. MANUSCRIPT ORGANIZATION AND FORMATTING:

Sincerely,

Reviewer #1 (Comments to the Authors (Required)):

The authors addressed many of the comments, although mostly in the rebuttal. The edits have clarified some of the questions, and as before, I am supportive of publishing the manuscript in principle, but a few concerns remain to be addressed.

- The abstract still gives the impression that the work was conducted in vivo, in mouse SVZ, in particular lines 30-33. This should be rephrased, to mention the model that is actually being used. I think the model used by the authors is strong enough, that they should not try to hide this away.

- Fig.1B: If the authors prefer not to normalise to the average of DIV0 and demonstrate the baseline variability, they should at least provide the separate WBs with corresponding quantifications in supplementary information. In particular for the blots quantified in Figure 1 and 3, which are key to the interpretations of the study.

- Fig4: If I understand correctly, ROIs were quantified, and then the number of cells per ROI counted, to represent each cell as a separate dot on the graphs? This means that, for example in Fig.4C DIV4, rather than having quantified several tens of cells, there were 6 ROIs that were quantified? Although I agree it is a good approach to integrate several cells (of the same identity) per ROI, I don't think this is a correct representation of the n-numbers quantified for these graphs, as the ROIs are the n and should be presented as such. In addition, I am not sure what is meant by 2-5 independent experiments. Does it mean that some DIVs or cell types were only quantified in a certain number of experiments? This should be clearly specified for each quantification.

As a suggestion, if the authors end up modifying this figure further, it might be helpful to not colour-code the morphological parameters in A in the same colours as the cell-type colours used B-F?

Response to Reviewer #1:

The authors addressed many of the comments, although mostly in the rebuttal. The edits have clarified some of the questions, and as before, I am supportive of publishing the manuscript in principle, but a few concerns remain to be addressed.

- The abstract still gives the impression that the work was conducted *in vivo*, in mouse SVZ, in particular lines 30-33. This should be rephrased, to mention the model that is actually being used. I think the model used by the authors is strong enough, that they should not try to hide this away.

We fully agree with the Reviewer's comment concerning the lack of mention in the Abstract of the model used (the neurosphere assay model, NSA). Therefore, we have rephrased the sentence in lines 30-33 to "Here, we evaluated mitochondrial properties throughout NSC differentiation and in lineage-specific cells. For this, we used the neurosphere assay model (NSA) to isolate, expand and differentiate mouse subventricular zone (SVZ) postnatal NSCs." (lines 29-32).

- **Fig.1B: If the authors prefer not to normalise to the average of DIV0 and demonstrate the baseline variability, they should at least provide the separate WBs with corresponding quantifications in supplementary information. In particular for the blots quantified in Figure 1 and 3, which are key to the interpretations of the study.**

Concerning reviewers' question about the variability of the data at DIV0 (baseline), we have addressed this concern by adding two supplementary images (Figure S8 and S9) to the manuscript. These figures display the separated Western Blots along with the quantification of each protein level, which has been normalized by GAPDH. These new figures were referenced in the Immunoblot section of Materials and Methods (lines 435 and 436) and the Figure legends were also added to the manuscript (lines 851-862).

- **Fig4: If I understand correctly, ROIs were quantified, and then the number of cells per ROI counted, to represent each cell as a separate dot on the graphs? This means that, for example in Fig.4C DIV4, rather than having quantified several tens of cells, there were 6 ROIs that were quantified? Although I agree it is a good approach to integrate several cells (of the same identity) per ROI, I don't think this is a correct representation of the n-numbers quantified for these graphs, as the ROIs are the n and should be presented as such. In addition, I am not sure what is meant by 2-5 independent experiments. Does it mean that some DIVs or cell types were only quantified in a certain number of experiments? This should be clearly specified for each quantification.**

The Reviewer has raised an important question regarding the interpretation of the n-numbers quantified in graphs of Figure 4. Indeed, the number of cells per ROI was counted to represent each cell as a separated dot on the graph. As mentioned by the reviewer, in Fig 4C DIV4 in astrocytes, 6 ROIs was quantified. To be clear, we have included in the legend of Fig4 (lines 773-777) and FigS3 (lines 829-831) the number of biological replicates, which corresponds to the number of cells analysed (N), and the number of ROIs quantified (n).

Part of the legend of the Fig4 that was updated:

N=40/40/40 and n=6/7/13 (left to right on astrocytes' plot); N=40/40/43/45 and n=6/9/12/22 (left to right on neurons' plot); N=40/40 and n=5/7 (left to right on OPCs' plot); N=36/4/26/14 and n=33/4/26/13 (left to right on oligodendrocytes' plot). N represents the number of biological replicates and n corresponds to the number of ROIs analysed.

Part of the legend of the FigS3 that was updated:

N=40/40 and n=6/6 (left to right on NSCs' plot).

N represents the number of biological replicates and n corresponds to the number of ROIs analysed.

As a suggestion, if the authors end up modifying this figure further, it might be helpful to not colour-code the morphological parameters in A in the same colours as the cell-type colours used B-F?

We totally agree with the Reviewer's comment regarding the necessity for the colour-code of the morphological parameters in A to be distinct from those in the graphs. Therefore, we have changed the colours in Fig4 A accordingly.

April 9, 2024

RE: Life Science Alliance Manuscript #LSA-2023-02473RR

Prof. Vanessa A. Morais
Instituto de Medicina Molecular, Lisboa
Avenida professor Egas Moniz
Lisboa 1649-028
Portugal

Dear Dr. Morais,

Thank you for submitting your Research Article entitled "Lineage-specific changes in mitochondrial properties during neural stem cell differentiation". It is a pleasure to let you know that your manuscript is now accepted for publication in Life Science Alliance. Congratulations on this interesting work.

DISTRIBUTION OF MATERIALS:

Again, congratulations on a very nice paper. I hope you found the review process to be constructive and are pleased with how the manuscript was handled editorially. We look forward to future exciting submissions from your lab.

Sincerely,
